# CCR9/CXCR5 Co-Expressing CD4 T Cells Are Increased in Primary Sjögren’s Syndrome and Are Enriched in PD-1/ICOS-Expressing Effector T Cells

**DOI:** 10.3390/ijms241511952

**Published:** 2023-07-26

**Authors:** Anneline C. Hinrichs, Aike A. Kruize, Floris P. J. G. Lafeber, Helen L. Leavis, Joel A. G. van Roon

**Affiliations:** 1Department of Rheumatology & Clinical Immunology, University Medical Center Utrecht, Utrecht University, 3508 Utrecht, The Netherlands; 2Center for Translational Immunology, University Medical Center Utrecht, Utrecht University, 3508 Utrecht, The Netherlands

**Keywords:** Sjögren’s syndrome (pSS), CXCR5/CCR9 T cells, Tph, Tfh, ICOS, PD-1

## Abstract

Primary Sjögren’s syndrome (pSS) is an autoimmune disease characterised by B cell hyperactivity. CXCR5+ follicular helper T cells (Tfh), CXCR5-PD-1hi peripheral helper T cells (Tph) and CCR9+ Tfh-like cells have been implicated in driving B cell hyperactivity in pSS; however, their potential overlap has not been evaluated. Our aim was to study the overlap between the two CXCR5- cell subsets and to study their PD-1/ICOS expression compared to “true” CXCR5/PD-1/ICOS-expressing Tfh cells. CXCR5- Tph and CCR9+ Tfh-like cell populations from peripheral blood mononuclear cells of pSS patients and healthy controls (HC) were compared using flow cytometry. PD-1/ICOS expression from these cell subsets was compared to each other and to CXCR5+ Tfh cells, taking into account their differentiation status. CXCR5- Tph cells and CCR9+ Tfh-like cells, both in pSS patients and HC, showed limited overlap. PD-1/ICOS expression was higher in memory cells expressing CXCR5 or CCR9. However, the highest expression was found in CXCR5/CCR9 co-expressing T cells, which are enriched in the circulation of pSS patients. CXCR5- Tph and CCR9+ Tfh-like cells are two distinct cell populations that both are enriched in pSS patients and can drive B cell hyperactivity in pSS. The known upregulated expression of CCL25 and CXCL13, ligands of CCR9 and CXCR5, at pSS inflammatory sites suggests concerted action to facilitate the migration of CXCR5+CCR9+ T cells, which are characterised by the highest frequencies of PD-1/ICOS-positive cells. Hence, these co-expressing effector T cells may significantly contribute to the ongoing immune responses in pSS.

## 1. Introduction

Primary Sjögren’s syndrome (pSS) is a systemic autoimmune disease characterised by lymphocytic infiltration of exocrine glands and B cell hyperactivity as a hallmark of the disease. Increased B cell activity is often reflected by glandular B cell infiltration and increases in local IgM and IgG-producing plasma cells, autoantibody production, systemic hypergammaglobulinemia and, in 5–10% of patients, B cell lymphoma development [1,2,3,4].

Several CD4 T cell subsets have been described that can contribute to B cell hyperactivity. Follicular helper T cells (Tfh) are C-X-C chemokine receptor 5-expressing (CXCR5+) CD4 T cells that were shown to be key drivers of B cell hyperactivity in many inflammatory conditions, including pSS [5,6,7,8]. In more recent years, several CD4 T cell subsets that lack CXCR5 expression (CXCR5-), such as peripheral T helper cells (Tph) and C-C motif chemokine receptor 9-expressing (CCR9+) follicular helper-like T cells (CXCR5-CCR9+ T cells or CCR9+ Tfh-like cells), have also been described with potent B-cell-activating features [9,10]. These cells were found to have Tfh-like activities in several inflammatory diseases including autoimmune diseases such as rheumatoid arthritis (RA) and systemic lupus erythematosus (SLE) [9,11]. The present study aimed to study the frequencies and phenotypic features of these three B-cell-activating cell types to reveal potential commonalities and dissimilarities.

Tfh cells are canonically defined as memory CD4 T cells, with expression of transcription factor Bcl6, surface expression of CXCR5, ICOS and PD-1 and secretion of IL-21 [12,13]. Even though cells defined in this way best resemble activated Tfh cells found in lymphoid structures, in some studies—including those studying Sjögren’s syndrome—(memory) CXCR5+ CD4 T cells without extensive further phenotyping are pragmatically also considered Tfh cells [7,14]. Elevated frequencies of circulating PD-1/ICOS/CXCR5-expressing Tfh cells have been reported in pSS patients and were associated with increased B cell activity and autoimmunity in these patients [5,15,16,17,18,19]. In the salivary glands of pSS patients, CXCR5, CXCR5-expressing CD4 T cells and the chemokine specifically mediating migration of CXCR5-expressing cells to the glands, CXCL13, are overexpressed [7,20,21,22,23,24,25,26,27]. Increased expression of CXCL13 is associated with increased lymphoid aggregates (foci) and increased organisation into ectopic lymphoid structures (ELS) and is subsequently associated with increased B cell hyperactivity and lymphoma development [26,27,28,29]. PD-1/ICOS/CXCR5-expressing Tfh cells, epigenetically quantified, were found to robustly correlate with lymphocytic focus scores, CXCL13 expression and B cell hyperactivity [24].

B-cell-activating Tph cells are defined as CXCR5-PD-1^hi^ memory CD4 T cells and have been found to be increased in the blood of pSS patients [16,30,31] These cells do not express Bcl6, are PD-1^hi^, can co-express ICOS and secrete high levels of CXCL13 and IL-21 [9,11,32]. In pSS patients, Tph cells in peripheral blood are more abundant than in healthy controls (HC) and a correlation was found between CXCR5-PD-1^hi^ cells (not gated on memory) and plasmablasts and rheumatoid factor positivity [31].

CXCR5-CCR9+ Tfh-like cells are potent B-cell-activating cells that produce high amounts of e.g., IL-21 and IFN-γ [10,33]. pSS patients have a higher frequency of CCR9+ Tfh-like cells in their circulation and these cells frequently demonstrate an effector phenotype, e.g., with elevated PD-1/ICOS expression, high CD127 (IL-7Rα) expression and IFN-γ and CCL5 production [10,34]. In addition, in the salivary glands of pSS patients, numbers of CCR9+ CD4 T cells are elevated, and CCL25, the cytokine mediating chemotaxis of CCR9+ cells, is overexpressed [10]. Increased CCL25 levels in secretomes of salivary gland tissue have been associated with autoimmunity (SSA positivity) and increased B cell hyperactivity (serum IgG levels) and levels of IL-21 and soluble IL-7R [25].

It is unknown to what extent overlap between Tfh, Tph and CCR9+ Tfh-like cells exists. Also, the expression of typical Tfh/Tph/CCR9+ Th cell-associated markers PD-1 and ICOS in these cell subsets, including CXCR5/CCR9 co-expressing cells, has not been simultaneously studied for all of these cell subsets. In this study, we first evaluated the overlap of CCR9+ Tfh-like cells and Tph cells. Then, we compared the expression of hallmark activation markers PD-1 and ICOS in the four different CXCR5/CCR9-defined cell subsets (Tfh, Tph, CCR9+ Tfh-like cells and CXCR5/CCR9 co-expressing cells) in pSS patients and HC.

## 2. Results

In this study, we defined Tph cells as CXCR5-PD-1^hi^ memory (CD45RO+), CCR9+ Tfh-like as CXCR5-CCR9+ and CCR9+ Tfh as CXCR5+CCR9+ memory CD4 T cells, as previously described (refs). CXCR5+CCR9+ co-expressing cells were not specified on the memory phenotype. Tfh cells were defined as CXCR5+ memory and “true” Tfh cells as CXCR5+PD-1+ICOS+ memory cells.

### 2.1. CXCR5-PD-1^hi^ Tph Cells and CXCR5-CCR9+ Tfh-like Cells Are More Abundant in pSS Patients but These Cell Types Show Little Overlap in Blood

From 12 female pSS patients and 11 age-matched female HC, Tph cells and CCR9+Tfh-like cells were studied. In pSS patients, the number of Tph cells was significantly increased compared to HC (median 0.53% versus 0.23% of CD4 T cells, *p* = 0.049) (Figure 1A). Also, the number of CCR9+ Tfh-like cells was significantly elevated in pSS patients compared to HC; i.e., approximately 2.7% in pSS patients versus 1.7% in HC (medians, *p* = 0.019) (Figure 1B). CCR9 expression was equally distributed between pSS patients and HC in Tph cells. Only a modest number of 2.1% (1.2–3.2%) of Tph cells expressed CCR9 (median with interquartile range) (Figure 1C). Enrichment of CCR9 in PD-1^hi^ and PD-1int compared to PD-1- was observed (Appendix A). In summary, from CD4 T cells in the peripheral blood of both pSS patients and HC, 0.011% (0.007–0.019%) expressed CD45RO+CXCR5-CCR9+PD-1^hi^ (median with interquartile range, Figure 1D). Of this specialised cell subset, a trend was seen for higher abundance in pSS patients (median values 0.009% and 0.013% for HC and pSS, respectively, *p* = 0.12).

### 2.2. PD-1^hi^ Expression Is Highest in CXCR5+ Memory Cells, CCR9 and CXCR5/CCR9 Co-Expressing Memory PD-1^hi^ Cells Show Increased ICOS Expression

After having determined that Tph cells and CCR9+ Tfh-like cells have limited overlap and that Tph cells are mainly CXCR5-CCR9-PD-1^hi^ memory cells, we compared the number of PD-1^hi^ cells among all memory subsets and the expression of ICOS within PD-1^hi^ memory cell subsets.

The percentage of PD-1^hi^ memory cells was not different in pSS patients compared to HC. CXCR5+ (Tfh) cells are characterised by the highest frequency of PD-1^hi^ cells, either with or without CCR9 co-expression (Figure 2A, PD-1^hi^ expression did differ between CXCR5+ T cell subsets compared to CXCR5-CCR9- and CXCR5-CCR9+ T cells, both *p* < 0.0001.) No difference in PD-1^hi^ expression was seen between CXCR5+CCR9- and CXCR5+CCR9+ cells (*p* = 0.57).

Within the PD-1^hi^ populations, cells expressing either CXCR5 and/or CCR9 of pSS patients showed a trend of higher ICOS expression compared to HC but this was only significant in CCR9+ Tfh-like cells (28.6% and 55%, medians for HC and pSS, respectively, *p* = 0.02, Figure 2B). Taking into consideration that CCR9+ Tfh-like cells have a low percentage of PD-1^hi^ cells, this implicates that the ratio between PD-1^hi^ and ICOS is quite different in these cells compared to CXCR5+ cells (Figure 2B, Appendix A).

Given the pronounced upregulation of ICOS on CCR9+ Tfh-like cells, we next compared ICOS expression between the different PD-1^hi^ cell subsets compared to CXCR5-CCR9- cells. Cells that expressed either chemokine receptor CXCR5 or CCR9 had a higher percentage of ICOS-expressing cells in their PD-1^hi^ population than CXCR5-CCR9- CD4 T cells. A trend for the highest ICOS expression within the PD-1^hi^ subset was seen for CXCR5/CCR9 co-expressing cells (CCR9+ Tfh) in pSS patients (*p* = 0.08 and *p* = 0.06 compared to CXCR5+CCR9- and CXCR5-CCR9+ cells, respectively) (Figure 2C). Significantly increased ICOS expression in HC was only seen for CXCR5-expressing Tfh cells compared to CXCR5-CCR9- cells (Appendix A).

### 2.3. CXCR5/CCR9 Co-Expressing Memory and Effector Cells Are Enriched for ICOS+PD-1+ Cells and Are Enriched in pSS

So far, in the above-mentioned analyses we adhered to the Tph definition of PD-1^hi^ memory cells. However, we next studied the expression of ICOS and PD-1 in CXCR5/CCR9-defined cells, since PD-1 expression has more frequently been used as a marker in cell phenotyping than PD-1^hi^ expression. For this, we focused on memory cells to allow comparison of the CCR9 and CXCR5-defined subsets. In fact, memory and effector cell populations in our study showed an overall significantly increased PD-1 and ICOS expression compared to naive cells (Appendix A). While there was a trend for increased ICOS and PD-1 expression in pSS patients compared to HC for CXCR5-expressing cells, this was only significant for PD-1/ICOS co-expression in CXCR5+CCR9- cells and ICOS expression of CXCR5+CCR9+ co-expressing cells (Figure 3A). When pooling patients and controls, in memory cells of the four different CXCR5/CCR9-defined subsets, CXCR5-CCR9- (“double negative”—DN) cells showed the lowest expression of PD-1 and ICOS+PD-1+ (compared to the three other subsets). ICOS expression was only significantly different in CXCR5+CCR9- cells compared to DN cells (Figure 3A). Furthermore, we wanted to evaluate if a difference was seen between the two Tfh cell subsets (i.e., CCR9- and CCR9+ memory CXCR5+ cells). PD-1, ICOS and PD-1/ICOS co-expression was significantly higher in CCR9+ Tfh cells than in CCR9- Tfh cells (in all three comparisons *p* < 0.05, Figure 3A).

As we established that higher expression of PD-1/ICOS was found in CXCR5/CCR9 co-expressing cells, we next evaluated if CXCR5+CCR9+ cell numbers were different in pSS patients compared to HC. Indeed, CXCR5+CCR9+ CD4 T cells were more abundant in pSS patients than in HC (Figure 3B). In addition, the number of true Tfh cells (CXCR5+ICOS+PD-1+ memory CD4) was also increased in pSS patients compared to HC (*p* = 0.03, Figure 3B). Within these true Tfh cells, 3.6% (2.9–6.8%) CCR9+ cells were observed (median with interquartile range). Calculated as percentages of CD4 T cells, a trend was seen for higher CCR9-expressing true Tfh cells in pSS patients compared to HC (*p* = 0.13, Figure 3B).

In earlier work from our group, CXCR5/CCR9-defined cells were not predefined in the memory phenotype [10,34]. It was found that approximately up to 10% of CCR9+ Tfh-like cells has an effector phenotype (CD27-CD45RO-) [10]. Given the potentially important role of such cells in immunopathology, we also studied PD-1/ICOS expression within this effector subset. Similarly to CD45RO+ memory cells, from the four CXCR5/CCR9-defined cell types, CXCR5-CCR9- effector cells show the lowest expression and CXCR5+CCR9+ cells show the highest expression of PD-1/ICOS (Figure 3C).

### 2.4. Increased Numbers of CCR9+ Tfh-like Cells and True Tfh Cells Are Associated with Autoimmunity

Finally, we evaluated if the four main cell subsets (Tph cells, true Tfh cells, CCR9+ Tfh-like cells and CXCR5+CCR9+ T cells) might be associated with B cell hyperactivity or are associated with clinical parameters (ESSDAI, LFS, percentage IgA in minor salivary gland biopsy). In this small study population, we did not find significant correlations between any of the subsets with ESSDAI or local inflammatory parameters. The percentage of both true Tfh cells and of CCR9+ Tfh-like cells from CD4 T cells was significantly higher in pSS patients with anti-SSA antibodies compared to HC (Figure 4).

In the present study, three patients took medication whereas nine did not. For all the analyses performed, no significant differences were observed. For CCR9+ T cells (2.57 ± 0.45 vs 2.71 ± 0.06, *p* = 0.864), Tph cells (0.65 ± 0.38 vs 0.86 ± 0.80, *p* = 0.864) and CCR9+CXCR5+ co-expressing Th cells (0.42 ± 0.17 vs 0.64 ± 0.24, *p* = 0.497), without and with medication, respectively, no significant differences were observed.

## 3. Discussion

In this study, we evaluated the frequency and phenotype of CCR9+ Tfh-like cells and Tph cells in peripheral blood to study the overlap between these populations in pSS. Typical markers associated with Tfh activity, PD-1 and ICOS, were assessed in these two CXCR5- cell subsets and in CXCR5+ Tfh cells. Our results show that the overlap between both memory CXCR5- populations is limited. From four CXCR5/CCR9-defined populations, PD-1^hi^ expression was most pronounced in CXCR5+ cells, whereas ICOS expression is markedly higher in both CXCR5 and CCR9 single-positive and double-positive cells. Furthermore, we showed that CXCR5+CCR9+ CD4 T cells are more abundant in pSS patients compared to HC and these cells are enriched in PD-1/ICOS expression in both the memory and effector cell subsets. Finally, we demonstrated that cell numbers of CCR9+ Tfh-like cells and true CXCR5+ memory PD-1+ICOS+ cells are increased in pSS patients with anti-SSA antibodies when compared to HC.

The minimal overlap between Tph cells and CCR9+ Tfh-like cells was not anticipated, since earlier work from our group showed that CCR9+ Tfh-like cells express upregulated PD-1 levels, especially in pSS patients [10]. In addition, both cell subsets share B-cell-stimulating functional properties and are characterised by the enrichment of IL-21-secreting cells and elevated expression of ICOS, CCR5 and CD127 [9,10,34,35]. This suggests that while functional properties may overlap, the chemokine receptor profile that CCR9+ cells acquire identifies them as a separate population, indicating that previous reports studying Tph or CCR9+ cells have likely studied separate populations. The expression of CCL25 by inflamed epithelial cells could direct CCR9+ cells to the inflamed site [36]. CXCL13 expressed in, e.g., germinal centres (GCs) or ectopic lymphoid structures (ELS) directs migration of CXCR5+ cells towards these specialised locations in the glands [26,27].

It is unclear how the chemotaxis of Tph cells is regulated but on an RNA level, increased expression of CCR2, CCR3, CCR5, CXCR3, CX3CR1 and CXCR6 in Tph cells compared to CXCR5+ and PD-1- cells has been reported [9]. Ligands for several of these chemokine receptors, e.g., CCL5, CXCL10 and CX3CL1 (fractalkine), are elevated in pSS patient’s salivary glands [37,38,39]. Hence, it is very likely that these overexpressed chemokines can facilitate the migration of Tph cells to the site of inflammation and contribute to immunopathology given their activated status.

Despite the fact that the CD4 T cell subsets described in the present study might be directed to specific areas in the inflamed tissues in pSS, they share functional properties such as ICOS and IL-21 expression. Hence, inhibiting these cell types might be achieved by targeting common surface-expressed proteins such as ICOS. In fact, all cell types studied here can be targeted by blocking ICOS/ICOSL interactions. Cells with ICOS expression can interact with cells expressing ICOSL (e.g., activated B cells). By blocking this pathway, the stimulatory signal could be blocked, resulting, e.g., in less release of IL-21 and the blockade of activation and further differentiation of Tph, Tfh and CCR9+ Tfh-like cells, thereby preventing B cell hyperactivity [40,41,42].

Our results in pSS fit the RNA sequencing results from circulating memory CD4 T cells in RA patients (sorted based on PD-1^hi^ versus PD-1- and CXCR5 expression), which indicates that the genes for ICOS and PD-1 (ICOS and PDCD1 genes, respectively) were upregulated most strongly in CXCR5+PD-1^hi^ cells [9]. Our results show that in PD-1^hi^ cells from pSS patients, PD-1/ICOS protein expression is the highest in CXCR5+CCR9+ Tfh cells. Of note, for CCR9 gene expression, a similar pattern was found with RNA sequencing, being most upregulated in CXCR5+PD-1^hi^ cells [9]. This is in line with earlier findings from our group showing that ICOS and PDCD1 gene expression are the highest in CXCR5+CCR9- CD4 T cells and the lowest in CXCR5-CCR9-T cells [34] Unfortunately, CXCR5+CCR9+ T cells were not studied here.

In the present study, we confirmed the findings of earlier studies showing that the abundance of Tph cells, CCR9+ Tfh-like cells and true Tfh cells are elevated in pSS patients compared to HC. In addition, we showed that the number of CD4 T cells co-expressing CXCR5/CCR9 is elevated in pSS patients. These CXCR5+CCR9+ T cells are enriched for PD-1/ICOS expression, particularly effector cells. Both CXCL13 and CCL25 are overexpressed in pSS and associated with B cell hyperactivity, the presence of autoantibodies and inflammatory mediators such as IL-21 and IFN-γ [10,26,27]. Given the tight connection between PD-1/ICOS co-expression and effector functions such as IL-21 secretion, this seems to indicate that even though these CXCR5/CCR9 chemokine receptor co-expressing cells form a small subset, these cells have very potent B-cell-activating potential. Their chemokine receptor co-expression indicates that these cells are primed to traffic to inflamed glands and to interact with B cells in GCs. Future research should further determine the functional properties of CXCR5+CCR9+ cells, preferably both in matched samples from circulation and salivary gland tissue.

Despite the implications for further research, our study does have some limitations. As a study that started out as an evaluation of the overlap between Tph cells and CCR9+ Tfh-like cells, we included a small study population, indicating that definite conclusions about associations with clinical features are difficult to draw. In addition, we only performed a phenotypic evaluation of peripheral blood samples and did not evaluate tissue samples. Nonetheless, we and others in previous studies have indicated enrichment of CCR9, CXCR5 and PD-1-expressing Th cells at inflammatory sites, including the inflamed glands of pSS patients [9,10,24,33]. Given the fact that the circulating cell populations we studied largely included memory/effector T cells that are known to home to inflammatory sites, our data might give important clues for their role in pSS immunopathology.

In conclusion, our study demonstrates little overlap between Tph cells and CCR9+ Tfh-like cells and that CXCR5+CCR9+ memory and especially effector CD4 T cells are enriched in PD-1/ICOS expression. Besides Tfh cells and the two more recently described CXCR5- cell subsets, i.e., Tph cells and CCR9+ Tfh-like cells, these CXCR5/CCR9 co-expressing cells may play a significant role in B cell hyperactivity in pSS.

## 4. Materials and Methods

### 4.1. Participant Inclusion

Peripheral blood mononuclear cells (PBMCs) were collected from n = 12 pSS patients and n = 11 age and sex-matched healthy controls (HC). All pSS patients were diagnosed by a rheumatologist and met the 2016 ACR-EULAR criteria [43]. All participants were included in the University Medical Center Utrecht (UMC Utrecht). The Medical Research Ethics Committee (METC) of the UMC Utrecht approved the study (reference number 13/697). All participants gave written informed consent. An overview of the demographic and clinical data are shown in Table 1.

### 4.2. Flow Cytometry

Fresh PBMCs from lithium-heparinised blood were isolated using Ficoll density gradient centrifugation. After collection, PBMCs were frozen and stored in liquid nitrogen until further use. PBMCs were thawed according to protocol and then stained with fixable viability dye eF780 (eBioscience) for 10 min at 4 °C. Then, the cells were stained with fluorochrome-conjugated antibodies against CD3, CD4, CD8, CCR9, CXCR5, PD-1, ICOS, CD27 and CD45RO for 25 min at 4 °C (details of the antibodies used can be found in Appendix A). Fluorescence minus one (FMO) controls were taken along to check marker expression. All samples were acquired on a BD LSRFortessa (BD Biosciences) using BD FACSDiva software v.8.0.1. (BD Biosciences). FlowJoTM v10.8 Software (BD Life Sciences) was used for data analysis.

### 4.3. Gating of PD-1^hi^ Cells

For flow cytometric analysis of Tph cells, memory CD4 T cells were gated on PD-1^hi^. The PD-1^hi^ gate was determined based on the histogram of PD-1 staining in memory versus naïve T cells combined with gating based on the PD-1/ICOS plot in memory CD4 T cells (Appendix A). This resulted in PD-1^hi^ gating in accordance with the gating previously described by Rao et al. with approximately 1% of memory PD-1^hi^ CD4 T cells in HC) [9,35].

## Figures and Tables

**Figure 1 ijms-24-11952-f001:**
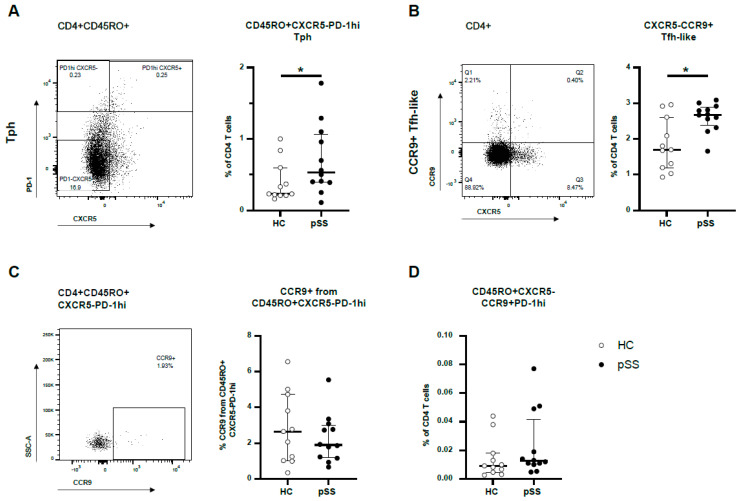
Tph and CCR9+ Tfh-like cells in pSS patients and HC showed minimal overlap. Flow cytometric analysis of peripheral blood mononuclear cells (PBMCs). (**A**) Representative dot plot and quantification of circulating Tph cells (CXCR5−PD-1hi memory CD4 T cells) in pSS patients and HC as percentages of CD4 T cells. (**B**) The frequency of CCR9+ Tfh-like cells in CD4 T cells in pSS patients and HC is shown. (**C**) The percentage of CCR9-expressing cells within Tph cells is depicted in a representative plot and quantified for all donors. (**D**) CXCR5−CCR9+PD-1hi memory cells expressed as the frequency of total CD4 T cell population in pSS patients and HC. Plots show medians plus interquartile ranges. Memory is defined as CD45RO+. HC: healthy control; pSS: primary Sjögren’s syndrome. Tph: T peripheral helper cell (memory CXCR5-PD-1hi CD4 T cell); CCR9+ Tfh-like: C-C motif chemokine receptor 9-expressing T follicular helper-like cell (CXCR5−CCR9+ CD4 T cell) * *p*-value < 0.05.

**Figure 2 ijms-24-11952-f002:**
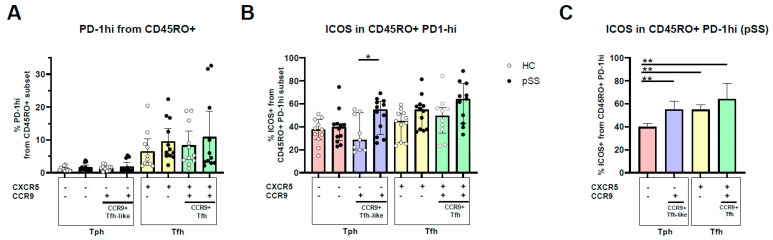
CCR9 and CXCR5/CCR9 co-expressing memory PD-1hi cells displayed increased ICOS expression. (**A**) The frequency of PD-1hi-expressing cells in the memory (CD45RO+) compartment of CXCR5/CCR9-defined cell subsets for pSS patients and HC. (**B**) ICOS expression is shown in PD-1hi memory subsets for pSS patients and HC. (**C**) Comparison of ICOS expression between the memory PD-1hi CXCR5/CCR9-defined subsets in pSS. Plots show medians plus interquartile ranges. HC: healthy control; pSS: primary Sjögren’s syndrome. Tph: T peripheral helper cell (memory CXCR5−PD-1hi CD4 T cell); CCR9+ Tfh-like: C-C motif chemokine receptor 9-expressing T follicular helper-like cell (CXCR5−CCR9+ CD4 T cell); ICOS: inducible T cell co-stimulator; PD-1: programmed death-1; *, ** indicates *p*-value < 0.05, 0.01, respectively.

**Figure 3 ijms-24-11952-f003:**
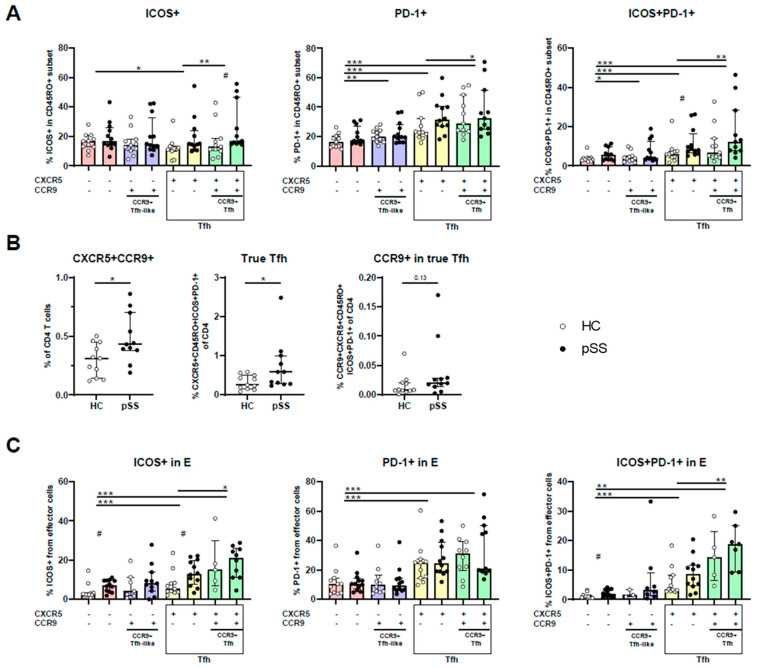
CXCR5/CCR9 co-expressing memory and effector cells are enriched in PD-1+ICOS+ cells. (**A**) Expression of ICOS, PD-1 and PD-1+ICOS+ in memory (CD45RO+) CXCR5/CCR9-defined cell subsets in pSS patients and HC, comparing the CXCR5−CCR9− (“double negative”—DN) subset to the three other subsets and the CXCR5+CCR9+ (“double positive”—DP) subset to the CXCR5+CCR9− subset. (**B**) Abundance of CXCR5+CCR9+ T cells, true Tfh cells and CCR9-expressing true Tfh cells from CD4 total in pSS patients and HC. (**C**) Expression of ICOS and PD-1 by CD27CD45RO- effector cell subsets in pSS patients and HC, comparing the DN subsets with the other subsets and the DP subset with the CXCR5 single-positive subset. Plots show medians plus interquartile ranges. HC: healthy control; pSS: primary Sjögren’s syndrome. CCR9+ Tfh-like: C-C motif chemokine receptor 9-expressing T follicular helper-like cell (CXCR5−CCR9+ CD4 T cell); ICOS: inducible T cell co-stimulator; PD-1: programmed death-1; *, **, *** indicates *p*-value < 0.05, 0.01, 0.001, respectively. # *p*-value < 0.05 in comparison pSS vs HC (Mann–Whitney test).

**Figure 4 ijms-24-11952-f004:**
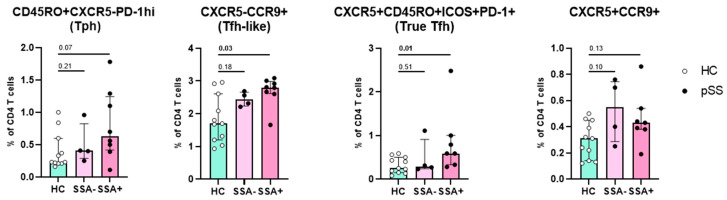
Association of Tph, Tfh-like, true Tfh and CCR9+ Tfh cell subsets with anti-SSA antibodies. Using PBMCs for flow cytometric analysis, the abundance of each cell subset from CD4 T cells is depicted. HC in green, pSS patients in pink. Anti-SSA negative patients (SSA−) in light pink and anti-SSA positive patients (SSA+) in dark pink. Plots show medians (interquartile range). HC: healthy control; pSS: primary Sjögren’s syndrome; anti-SSA: anti-Sjögren’s syndrome related antigen A antibody/anti-Ro. *p*-values shown are results of Wilcoxon nonparametrical paired test with significant differences in bold.

**Table 1 ijms-24-11952-t001:** Participants’ characteristics.

	HC (n = 11)	pSS (n = 12)
Female, n (%)	11 (100)	12 (100)
Age, years	55 (49–60)	59 (54–64)
Anti-Ro/SSA positive, n (%)		8 (67)
Anti-La/SSB positive, n (%)		4 (33)
ANA positive, n (%)		7 (58)
Lymphocytic focus score (foci/4 mm^2^)		1.8 (1.1–2.4)
IgA-positive plasma cells (%)		65 (40–81)
Schirmer (mm/5 min)		0 (0–1)
Serum IgG (g/L)		13.6 (11.4–15.2)
ESSDAI score (0–123)		4 (2–5)
ESSPRI score (0–10)		6.3 (5.3–7.3)
Immunosuppressants, n		3
Hydroxychloroquine, n		2

Medians with interquartile range (Q1–Q3) are shown unless specified otherwise. HC: healthy controls; pSS: primary Sjögren’s syndrome; anti-Ro/SSA: anti-Ro/Sjögren’s syndrome related antigen A antibody; anti-La/SSB: anti-La/Sjögren’s syndrome related antigen B antibody; ANA: antinuclear antibody; ESSDAI: European League Against Rheumatism (EULAR) Sjögren’s syndrome disease activity score; ESSPRI: EULAR Sjögren’s syndrome patient reported index.

## Data Availability

The original contributions presented in the study are included in the article and Appendix A. Further inquiries can be directed to the corresponding author.

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
