# Peer review of "CCR9/CXCR5 Co-Expressing CD4 T Cells Are Increased in Primary Sjögren’s Syndrome and Are Enriched in PD-1/ICOS-Expressing Effector T Cells"

_ijms, 2023, doi:10.3390/ijms241511952_

Round 1
Reviewer 1 Report
In this study, the overlap of CCR9+ Tfh-like cells and Tph cells were first evaluated in the peripheral blood of pSS patients and HC. Then the expression of hallmark activation markers PD-1 and ICOS on the four different CXCR5/CCR9-defined cell subsets (Tfh, Tph, CCR9+ Tfh-like cells, and CXCR5/CCR9 co-expressing cells) was compared in the peripheral blood of pSS patients and HC. It is interesting work. However:
It will be more relevant to pSS pathogenesis if these studies would have been done on the lymphocytic infiltrates of the labial minor salivary glands of pSS patients and sicca controls. Furthermore:
It is not proper, in such studies, to include pSS patients on immunosuppressive medications (3/12) since immunosuppression may alter the blood lymphocytic composition. Why the 3 pSS patients were on immunosuppressive medications?
The spectrum of pSS studied is obscured by using the ESSDAI score. From the ESSDAI score of pSS patients studied it appears that this patient group had a mild disease. It will be of interest if the authors had applied parametric correlation among the individual ESSDAI score of each pSS patient and the percent of the cell population of the same pSS patient.
In Table 1 the ESSDAI score range is 1-12 and not 1-123!
General References on pSS are highly biased.
Author Response
We would like to thank the reviewer for his/her useful suggestions. They have been addressed point by point below.
Comments
1) It will be more relevant to pSS pathogenesis if these studies would have been done on the lymphocytic infiltrates of the labial minor salivary glands of pSS patients and sicca controls.
Response. We agree with the reviewer that it is very relevant to study the presence and activity of these T cells in the tissue. Unfortunately we did not (and do not) have the resources to analyze this for the current paper (importantly also due to COVID in the last years). Also, although we and others previously demonstrated increased CCR9 and CXCR5-expressing cells in the tissue of pSS patients, the number of CCR9 cells was very low. In vitro we have shown that upon activation CCR9 T cells rapidly lose expression of CCR9. In addition, upon encountering ligand CCL25, of which the expression is increased, receptor (CCR9) expression may be reduced. Given the low percentage of double expressors we therefore anticipate that it is difficult to study phenotypic and functional properties of CCR9/CXCR5-expressing cells in the tissue.
However, given the fact that the circulating cell populations we studied largely included memory/effector T cells that are known to home to inflammatory sites our data might give important clues for their role in pSS immunopathology. We have described this in the discussion
2) It is not proper, in such studies, to include pSS patients on immunosuppressive medications (3/12) since immunosuppression may alter the blood lymphocytic composition. Why the 3 pSS patients were on immunosuppressive medications?
Response. We understand the concern of the reviewer. However, for none of the studied parameters we have seen significant differences between those patients with and without treatment. Also please appreciate that in previous studies others and we have shown data that are in line with present data, confirming increased numbers of CCR9 and Tph CD4 T cells.
CXCR5- Tph cells and CCR9+ Tfh-like cells, both in pSS patients and HC showed limited overlap (total n=23). We showed robust increased PD-1/ICOS expression in memory cells expressing CXCR5 or CCR9. However, both for HC and pSS the highest expression was found in CXCR5/CCR9 co-expressing T cells. These features most likely are intrinsic to effector T cells and seem minimally affected by the medication in our study. Importantly, in line with single expressing CCR9 Th and Tph cells, co-expressing Tph cells are enriched in the circulation of pSS patients.
3) The spectrum of pSS studied is obscured by using the ESSDAI score. From the ESSDAI score of pSS patients studied it appears that this patient group had a mild disease. It will be of interest if the authors had applied parametric correlation among the individual ESSDAI score of each pSS patient and the percent of the cell population of the same pSS patient.
Response. We agree with the reviewer that this patient group had mild disease activity. Inclusion of patients by Dr Hinrichs occurred in the past years and has been dramatically hindered by the COVID pandemic. Surely we would have used increased numbers of patients, including high activity patients. However, we would like to point out that in high throughput analyses studying proteomics and transcriptomics of blood cells, correlations with ESSDAI scores are usually very low. Much better correlations are found with B cell hyperactivity, autoimmunity and IFN scores as partly shown in the present study. To illustrate this, based on proteomics, we recently described an inflammatory endotype within patients that had equal disease activity in terms of ESSDAI scores. These patients profoundly responded to Leflunomide/Hydroxychloroquine combination therapy (van der Heijden et al. Lancet Rheumatology 2020).
4) In Table 1 the ESSDAI score range is 1-12 and not 1-123!
Response. Please appreciate that this range is referring to the score range of the ESSDAI index itself, which theoretically can reach 123 points, but in daily practice and even upon selection of active patients for clinical trials is way lower. In the latter case inclusion starts when higher than 5 points and usually patients have average scores of around 10.
5) General References on pSS are highly biased.
Response. The reviewer does not indicate which bias has occurred. We objectively referred to studies underscoring our statements. We apologize if by accident bias has occurred and surely are willing to change this.
Reviewer 2 Report
The paper is interesting and well written. I suggest to discuss the role of Th17 cells in Sjogren syndrome (see and add as referecne paper by Murdaca et al concerning Th17 in chronic immunemediated diseases)
Minor english editing
Author Response
The paper is interesting and well written. I suggest to discuss the role of Th17 cells in Sjogren syndrome (see and add as referecne paper by Murdaca et al concerning Th17 in chronic immunemediated diseases)
Response. Thank for your kind words and interesting suggestion. However, In pSS Th1-driven immunopathology on different levels has been convincingly demonstrated, in contrast to Th17-induced immunopathology. Although the reviewer puts forward an interesting hypothesis whereby activated B cells play a role in subsequent T cell activation, in particular Th17 activation, this is speculative and needs quite some introduction and adaption to fit it in the manuscript. We hope the reviewer can appreciate this may be too speculative and beside the scope of the present study,
Reviewer 3 Report
In this article, Hinrichs et al analyzed CXCR5- Tph and CCR9+ Tfh-like cell populations from peripheral blood mononuclear cells of primary Sjogren's syndrome (pSS) patients and healthy controls (HC) were compared by flow cytometry. PD-1/ICOS expression from these cell subsets was compared to each other and to CXCR5+ Tfh cells, taking into account their differentiation status. CXCR5- Tph and CCR9+ Tfh-like cells are two distinct cell populations that both are enriched in pSS patients and can drive B cell hyperactivity in pSS. The known upregulated expression of CCL25 and CXCL13, ligands of CCR9 and CXCR5, at pSS inflammatory sites suggests concerted action to facilitate migration of CXCR5+CCR9+ T cells, which are characterised by the highest frequencies of PD-1/ICOS positive cells. They concluded that these co-expressing effector T cells may significantly contribute to the immune responses in pSS. The number of samples is small, but they are working on a detailed study regarding cell surface markers.
I have some questions below.
major concerns)
1) In table 1, background, there are several patients who already have these treatments in, but ideally you would need to examine them in untreated patients. If immunosuppressants or HCQs are in the background, they could affect the immune profile. It might be difficult because of the small number of patients, but please provide data on whether there is a difference in profiles with and without treatment.
2) Although the disease is different because it is not Sjögren's syndrome, it has been reported that B cells induce Th17 cells and Treg cells in systemic scleroderma (Fukasawa T, Yoshizaki A, Ebata S, Yoshizaki-Ogawa A, Asano Y, Enomoto A, Miyagawa K, Kazoe Y, Mawatari K, Kitamori T, Sato S. Single-cell-level protein analysis revealing the roles of autoantigen-reactive B lymphocytes in autoimmune disease and the murine model. Elife. 2021 Dec 2;10:e67209. doi: 10.7554/eLife.67209. PMID: 34854378; PMCID: PMC8639144.).In Discussion, you discuss the relationship between T cells and B cells. Please cite this paper and discuss the relationship between B cells and T cells.
minor concerns)
1) In line 374-383, supplementary materials and author contribution are still in the template and have not been updated. Please update them.
We can read English without any problems.
Author Response
We would like to thank the reviewer for his time and sharing his concerns. They have been addressed point by point below
1) In table 1, background, there are several patients who already have these treatments in, but ideally you would need to examine them in untreated patients. If immunosuppressants or HCQs are in the background, they could affect the immune profile. It might be difficult because of the small number of patients, but please provide data on whether there is a difference in profiles with and without treatment.
Response. We understand the concern of the reviewer. However, for none of the studied parameters we have seen significant differences between those patients with and without treatment. But please appreciate that for drawing reliable conclusions the group sizes need to be bigger. Also please appreciate that in previous studies others and we have shown data that are in line with present data, confirming increased numbers of CCR9 and Tph CD4 T cells in pSS patients.
CXCR5- Tph cells and CCR9+ Tfh-like cells, both in pSS patients and HC showed limited overlap (total n=23). We showed robust increased PD-1/ICOS expression in memory cells expressing CXCR5 or CCR9. However, both for HC and pSS the highest expression was found in CXCR5/CCR9 co-expressing T cells. These features most likely are intrinsic to effector T cells and seem minimally affected by the medication in our study. Importantly, in line with single expressing CCR9 Th and Tph cells, co-expressing Tph cells are enriched in the circulation of pSS patients.
2) Although the disease is different because it is not Sjögren's syndrome, it has been reported that B cells induce Th17 cells and Treg cells in systemic scleroderma (Fukasawa T, Yoshizaki A, Ebata S, Yoshizaki-Ogawa A, Asano Y, Enomoto A, Miyagawa K, Kazoe Y, Mawatari K, Kitamori T, Sato S. Single-cell-level protein analysis revealing the roles of autoantigen-reactive B lymphocytes in autoimmune disease and the murine model. Elife. 2021 Dec 2;10:e67209. doi: 10.7554/eLife.67209. PMID: 34854378; PMCID: PMC8639144.).In Discussion, you discuss the relationship between T cells and B cells. Please cite this paper and discuss the relationship between B cells and T cells.
Response. As the reviewer rightfully notices pSS and SSc are different diseases with different immunopathologies, where in pSS Th1-driven B cell hyperactivity and immunopathology on different levels has been convincingly demonstrated, in contrast to Th17-induced immunopathology. In SSc fibrotic events have been associated with Th2/Th17-induced immunopathology. Although the reviewer puts forward an interesting hypothesis whereby activated B cells play a role in subsequent T cell activation, this was not the subject of the present study. We hope the reviewer can appreciate this may be too speculative and beside the scope of this paper.
minor concerns)
1) In line 374-383, supplementary materials and author contribution are still in the template and have not been updated. Please update them.
Response. Thank you for noticing. We have changed that accordingly.
Round 2
Reviewer 3 Report
I am sorry, but I cannot say that the authors have adequately responded to the comments. The response says that there were no significant differences in parameters such as immunosuppressants, but no data is presented to support this. Therefore, the results may be misleading. It is also difficult to say that questions and suggestions have been adequately answered.
None.
Author Response
I am sorry, but I cannot say that the authors have adequately responded to the comments. The response says that there were no significant differences in parameters such as immunosuppressants, but no data is presented to support this. Therefore, the results may be misleading. It is also difficult to say that questions and suggestions have been adequately answered.
Response. We understand the concern of the reviewer. However, we tested 92 comparisons of the cell subsets described in the paper. In none of the comparisons between patients with or without medication there was a significant difference. To illustrate this we have added the data plus p-values of CCR9+ T cells, Tph cells and CCR9+ CXCR5+ co-expressing CD4 T cells with or without medication to the results section.
Round 3
Reviewer 3 Report
No additional comments.